Cytogenotoxic potential and toxicity in adult Danio rerio (zebrafish) exposed to chloramine T

Rivero-Wendt Carla Letícia Gediel 1
Miranda Vilela Ana Luisa 2
GarciaFernandes Luana 1
Negreli Santos Andreza 3
Leal Igor 3
Jaques Jeandre 3 4
Fernandes Carlos Eurico carlos.fernandes@ufms.br 1
1 Institute of Bioscience, Federal University of Mato Grosso do Sul, Laboratory of Experimental Pathology , Campo Grande , Mato Grosso do Sul , Brazil
2 Independent Research , Brasilia , Distrito Federal , Brazil
3 Institute of Bioscience, Federal University of Mato Grosso do Sul, Multicenter Graduate Program in Biochemistry and Molecular Biology , Campo Grande , Mato Grosso do Sul , Brazil
4 Faculty of Pharmaceutical Sciences, Food and Nutrition, Federal University of Mato Grosso do Sul, Graduate Program in Pharmaceutical Sciences , Campo Grande , Mato Grosso do Sul , Brazil
Oehlmann Jörg
Electronic publication date: 2023 Dec 4
Publication date: 2023
Volume: 11
Electronic Location ID: e16452
Received 2023 Jul 26; Accepted 2023 Oct 23
Copyright: ©2023 Rivero-Wendt et al.
Copyright year: 2023
Copyright holder: Rivero-Wendt et al.
License: This is an open access article distributed under the terms of the Creative Commons Attribution License, which permits unrestricted use, distribution, reproduction and adaptation in any medium and for any purpose provided that it is properly attributed. For attribution, the original author(s), title, publication source (PeerJ) and either DOI or URL of the article must be cited.
License URL: https://creativecommons.org/licenses/by/4.0/

Keywords: Integrated optical density (IOD), Nuclear abnormalities, Micronucleus test, Gill and liver histopathology, Acetylcholinesterase activity

Funding: The Coordenação de Aperfeiçoamento de Pessoal de Nível Superior - Brasil (CAPES) - Finance Code 001 The Fundação Universidade Federal de Mato Grosso do Sul –UFMS/MEC –Brazil Conselho Nacional de Desenvolvimento Científico e Tecnológico #310058/2010-1 This study was funded by the Coordenação de Aperfeiçoamento de Pessoal de Nível Superior - Brasil (CAPES) - Finance Code 001, and by the Fundação Universidade Federal de Mato Grosso do Sul –UFMS/MEC –Brazil; CEF has been continuously supported by Conselho Nacional de Desenvolvimento Científico e Tecnológico (grant #310058/2010-1). The funders had no role in study design, data collection and analysis, decision to publish, or preparation of the manuscript.

==============================
Background

Chloramine-T (CL-T) is a synthetic sodium salt used as a disinfectant in fish farms to combat bacterial infections in fish gills and skin. While its efficacy in pathogen control is well-established, its reactivity with various functional groups has raised concerns. However, limited research exists on the toxicity of disinfection by-products to aquatic organisms. Therefore, this study aims to assess the sublethal effects of CL-T on adult zebrafish by examining biomarkers of nucleus cytotoxicity and genotoxicity, acetylcholinesterase (AChE) inhibition, and histopathological changes.

Methods

Male and female adult zebrafish (wildtype AB lineage) specimens were exposed to 70, 140, and 200 mg/L of CL-T and evaluated after 96 h. Cytotoxic and genotoxic effects were evaluated by estimating the frequencies of nuclear abnormalities (NA), micronuclei (MN), and integrated optical density (IOD) of nuclear erythrocytes. Histopathological changes in the gills and liver were assessed using the degree of tissue changes (DTC). AChE activity was measured in brain samples.

Results and conclusions

At a concentration of 200 mg/L, NA increased, indicating the cytogenotoxic potential of CL-T in adult zebrafish. Morphological alterations in the nuclei were observed at both 70 and 200 mg/L concentrations. Distinct IOD profiles were identified across the three concentrations. There were no changes in AChE activity in adult zebrafish. The DTC scores were high in all concentrations, and histological alterations suggested low to moderate toxicity of CL-T for adult zebrafish.

Introduction

Disinfectants play a crucial role in various industries, including agriculture, fish farming, slaughterhouses, and kitchens (Haneke, 2022), by preventing the proliferation of bacteria, viruses, and fungi and maintaining water quality. When disinfectants are used for water disinfection, they react with natural organic matter, leading to the formation of disinfection by-products (DBPs), many of which are cytotoxic, genotoxic, mutagenic, and teratogenic in nature (Cui et al., 2021; Yadav, Dhodpakar & Kapley, 2020).

The use of disinfectants, such as Chloramine-T (CL-T), in fish farms is common to combat various pathogens that can impact fish production. For adult fish, typical CL-T concentrations for prophylaxis or disinfection between 5–20 mg/L are used. The baths can be used as static baths once a day for 60 min or as a low flow rinse treatment for three consecutive days (for intensive treatment) or alternate days with three baths of 60 min each (Powell, Speare & MacNair, 1994; Sanchez et al., 1996). However, their effects on non-target species and the environment can result in pollution of aquatic ecosystems. Thus, it is essential to control the use of chemical products to minimize environmental contamination (Agoba et al., 2017; Bahadir et al., 2019; Gustavino et al., 2005).

CL-T, a disinfectant employed since the 1990s for treating intensively cultured fish for human consumption (Bullock, Herman & Waggy, 1991), is a synthetic sodium salt known as N-chloro-r-toluenesulfonamide, with the chemical formula r-CH3C6H4SO2NClNa.3H2O (Nayak et al., 2022). It readily dissolves in water and is also used for chlorination in water treatment due to its reactivity with a broad range of functional groups. CL-T reacts with oxidizable materials, including amines, amino acids, humic substances, and other organic and inorganic compounds present in the treated water and subsequent dilution waters (Nayak et al., 2022; Schmidt et al., 2007).

Furthermore, it is important to consider (i) the continuous discharge of chlorine-treated waters in wastewater treatment plants; (ii) the intermittent discharge of aquaculture effluents containing CL-T in aquaculture facilities (averaging approximately 40 discharges per year per facility employing CL-T); and (iii) the scarcity of studies reporting the toxicity of DBPs to aquatic organisms (Schmidt et al., 2007).

Therefore, a comprehensive evaluation of the effects of CL-T on aquatic organisms is necessary to better understand its impact. This assessment would enable the investigation of ecotoxicological effects, examining both target and non-target species within fish farms and aquaculture systems.

In this context, freshwater aquatic organisms, such as Danio rerio (zebrafish), are used to assess the toxicity of chemicals. Zebrafish share 70–80% genetic homology with humans and possess vertebrate-like structures (Bertotto, Catron & Tal, 2020; Gunnarsson et al., 2008). Adult zebrafish and embryos serve as valuable model organisms for screening the toxicity of drugs and chemicals (Brannen et al., 2010). Recent studies have reported the lethality and sublethal toxicity associated with CL-T exposure using zebrafish embryos as model organisms. In a short-term acute toxicity test at 96 h, CL-T-induced abnormalities were observed in zebrafish embryos. Concentrations above 64 mg/L resulted in cardiac oedema and reduced heart rate. Furthermore, acetylcholinesterase (AChE) inhibition is consistent with the morphological and equilibrium disturbances observed. Therefore, the use of CL-T in zebrafish embryos should be used with caution and the effects on adults need to be assessed (Rivero-Wendt et al., 2023).

Therefore, to compare the findings obtained from zebrafish embryos and examine alternative biomarkers of effects that could potentially indicate threats to this species drove the objectives of this study. In the present study, our objective is to evaluate the sublethal effects of CL-T in adult zebrafish by examining biomarkers of nucleus cytotoxicity and genotoxicity, enzymatic inhibition of AChE, and histopathological changes.

Material & Methods

Experimental design

The zebrafish (wildtype AB lineage) used in this study were obtained from a quality-certified aquarium (Ogawa e Sato LTDA) acquired at 3 months of age, with a length of 2–3 cm and acclimated at the Aquaculture Station of the Federal University of Mato Grosso do Sul (CEUA establishment license no. 3.128/2021). They were housed in 10 L aquariums for the acclimatization period. In the test, a total of 40 specimens were used, males and females, 10 specimens were used per treatment (control, 70, 140 and 200 mg/L) and divided into three replicates. All samples were obtained from treated and control specimens of both sexes, with no recording of the sexual ratio. Only fish that survived until the end of the 96-hour exposure period were utilized. Two fish died when exposed to 140 mg/L and 200 mg/L concentrations, respectively. The water conditions were maintained with a waterfall filter, temperature of 28 °C, pH of 7 ± 0.5, conductivity of 55 ± 50 µS/cm, and dissolved oxygen levels equal to or above 95% saturation. A 12:12 h photoperiod cycle was followed. The adult zebrafish were fed twice daily with a commercial artificial diet (TetraMin flakes fish food, Germany). Sublethal concentrations of CL-T (C7H7CINNaO2S. 3H2O) (Halamid®) for adult zebrafish were determined based on the LC50 values obtained from previous evaluations in zebrafish embryos (143.05 ± 3.11 mg/L at 24 h and 130.97 ± 7.40 mg/L at 96 h) (Rivero-Wendt et al., 2023). CL-T (Halamid®) with 98% purity was obtained from Western Chemical Inc. Exposure to CL-T was static for 96 h. The animals were euthanized following Concea (2018). All experimental procedures were conducted following the guidelines and regulations approved by the Ethics Committee for the Experimental Use of Animals (CEUA; approval no. 3.128/2021).

Nuclear cytotoxicity/genotoxicity and integrated optical density

The methodology employed for the test followed the procedure outlined by Hooftman & de Raat (1982). After euthanasia, peripheral blood samples were collected from the caudal vein of the zebrafish using a 10 µL micropipette containing 3% EDTA. Immediately after collection, slide smears were prepared from the blood samples. The slides were fixed in methanol for 5 min, air-dried, and then stained with May-Grünwald-Giemsa-Wright (MGGW) solution using the modified Rosenfeld method to enhance the visualization of erythrocyte nuclei for morphological analysis (Ranzani-Paiva et al., 2013).

Nuclear abnormalities (NA) were defined according to Carrasco, Tilbury & Myers (1990). Erythrocytes with two nuclei were considered binuclear; blebbed nuclei, when they exhibited a relatively small evagination of the nuclear membrane, containing euchromatin; lobed nuclei, when evaginations larger than the blebbed nuclei, which can have several lobes; and vacuolated nuclei, when an appreciable depth chromatin failure without nuclear material was observed. In this work, one thousand erythrocytes cells with complete cytoplasm were scored per fish and specific NA types were not discriminated (Carrasco, Tilbury & Myers, 1990).

For the analysis of micronucleus (MN), three thousand erythrocyte cells with complete cytoplasm were scored per fish. MN were identified when they were smaller than one-third of the main nuclei (a), did not touch the main nuclei (b), and exhibited a non-refractive circular or ovoid chromatin body with a similar staining pattern as the main nucleus (c) (Rivero-Wendt et al., 2020).

To calculate the IOD of the erythrocyte nuclei, ten images (RGB 4,096 × 3,286 pixels) were randomly captured from each blood smear slide at a magnification of 1,000× using bright-field microscopy (Opticam 500R®) equipped with a LOPT1001® camera. The images were transformed into 8-bit type and the threshold tool in ImageJ software version 1.53a (Ferreira & Rasband, 2012) was used for image analysis. In the “Analyze/Set Measurements” menu, the area (µm2), circumference (µm), roundness (0–1), and Ferret’s diameter (µm) of 15 randomly selected nuclei per image were measured. The calculation of IOD followed the method described by Hardie, Gregory & Hebert (2002), as follows: ∑i=0n−log10IFiIBi

in which n = total number of pixels in the nucleus; IFi = intensity of the nuclear pixels; and IBi = intensity of background image pixel (Hardie, Gregory & Hebert, 2002).

Acetylcholinesterase (AChE) activity inhibitory assay

The inhibition of AChE was assessed following the Ellman’s assay protocol (Ellman et al., 1961). The primary modifications involved the use of a phosphate/HEPES buffer (50/5 mM, pH 7) to enhance the stability of 5,5-dithiol-bis-(2-nitrobenzoic acid) (DTNB), which was added after catalysis to minimize potential interference with AChE activity. Tetraisopropyl pyrophosphoramide (iso-OMPA) was employed as a selective inhibitor of butyrylcholinesterase (BuChE). After conducting the fish toxicity test, the brains of adult fish (n = 38) were crushed in 10 mM Tris buffer (pH 7.2) using a mortar and pestle to create homogenates, while keeping them on ice. The enzymatic activity analysis was performed in 96-well plates using the following experimental procedure, with a final volume of 300 µL. Briefly, 150 µL of phosphate/HEPES buffer (100/10 mM, pH 7), 80 µL of distilled water, 20 µL of homogenates, and 20 µL of 292 µM iso-OMPA were mixed. The plate was preincubated at 37 °C for 30 min. To initiate the reaction, 30 µL of 10 mM acetylthiocholine (ACSCh) was added and incubated at 37 °C for 30 min. Subsequently, 20 µL of 51 mM neostigmine bromide prepared in phosphate/HEPES buffer (100/10 mM, pH 7) was added to complete the reaction. Finally, 20 µL of 8.5 mM DTNB prepared in phosphate/HEPES buffer (100/10 mM, pH 7) was promptly added to induce the production of thiocholine. The hydrolysis of acetylthiocholine (ACSCh) by AChE leads to the formation of 5-thio-2-nitrobenzoate anion (TNB) through the reaction with DTNB. The concentration of TNB was determined at λmax = 412 nm using a SpectraMax Plus 384 Microplate Spectrophotometer (Molecular Devices LLC, USA) at room temperature. The activity of AChE was expressed as µmol hydrolyzed ACSCh/hr/mg protein. The protein content of the samples was determined using the method described by Bradford (1976) with bovine serum albumin (BSA) as the standard (Bradford, 1976).

Gills and liver histopathology

Following euthanasia, the head of the fish was separated from the body for immediately immersed in Davidson’s solution for 24 h and subsequently subjected to decalcification for three days using a solution containing 0.7 g EDTA, 8 g sodium and potassium tartrate, 0.14 g sodium tartrate, 120 mL HCl, and distilled H2O to a final volume of 900 mL. The fish specimens were then processed for histological analysis by embedding them in paraffin, cutting them into 3 µm-thick sections, and staining with hematoxylin and eosin for examination under bright-field microscopy (Suvarna, Layton & Bancroft, 2019).

Histopathological findings were classified based on the degree of tissue changes (DTC) as described by Bernet et al. (2001), which adopted the standard reaction (a) features and assigned importance scores (w) to these changes. Table 1 shows the histological alterations considered in this study and their corresponding importance degrees. The DTC was estimated using the following formula: (1) DTC=∑alta×w,

in which a represents the distribution of damage (0 = absent; 1 = minor; 2 = moderate; and 3 = marked occurrence) and w represents the reversibility degree of the damage (1 = easily reversible; 2 = moderate alterations with probable reversion after exposure; and 3 = irreversible alterations). The frequency (%) of each w value, adjusted according to the respective values, was calculated based on the overall sum (∑alt) for each alteration using the following formula: F% = [(a × w)/∑alt] ×100 (Silva et al., 2021).

Table 1 Degree tissue changes (DTC)* according to stages of lesions observed in liver and gill.

Liver	Stages	
Hemorrhage/hyperemia/congestion/edema sinusoidal/perivascular	1	
Cellular tumefaction/hydropic degeneration	1	
Cytoplasmic inclusion/hyaline/lipofuscin	1	
Melanomacrophages deposits	1	
Cell atrophy/hypertrophy	2	
Nuclear alterations (pyknosis/karyolysis/karyorrhexis/inclusions)	2	
Granulocytic or agranulocytic infiltrate	2	
Cell necrosis (focal/diffuse)	3	
Sinusoidal epithelial necrosis	3	
Gill		
Hypertrophy of lamellar epithelium	1	
Hyperplasia of lamellar epithelium	1	
Fusion of lamelar epithelium	1	
Mucous cell proliferation	1	
Edema of epithelial cell	1	
Shortening of secondary lamella	1	
Lamellar atrophy	1	
Adherence of lamelar epithelium	1	
Mucous cell hyperplasia and hypertrophy	1	
Epithelial lifting	1	
Vasodilatation and vascular hyperemia	1	
Leukocyte infiltrate	1	
Telangiectasis (aneurysm)	1	
Rupture of lamelar epithelium	2	
Lamellar thrombosis	2	
Epithelial fibrosis	3	
Fibrous lamellar thrombosis	3	
Necrosis	3	
Notes.

* According to Bernet et al. (2001); 1, easily reversible; 2, moderated alterations with probable reversion after end of exposure and 3, irreversible alterations.

Statistical analysis

The normal distribution of continuous variables was assessed using the Shapiro–Wilk test. Differences between the groups under analysis were evaluated using either ANOVA or the Kruskal–Walli’s test, depending on the data distribution. When ANOVA yielded significant results, the Bonferroni’s post-hoc test was used for pairwise comparisons between the treatments. For significant Kruskal-Wallis results, the Mann–Whitney U test was employed to examine differences between the treatments through pairwise comparisons.

Thus, the effect of CL-T concentrations on the erythrocyte MN test and histopathological findings DTC was compared using Kruskal–Wallis analysis of variance with the Mann–Whitney U test, as the data were not normally distributed. Total NA, erythrocyte nuclear morphometric data (area, circumference, roundness, and volume), and AChE activity inhibitory assay were compared using a one-way ANOVA (general linear model—GLM). The intensity over distance (IOD) curve fit was generated from the normalized values (y/mean). Box plots were used to represent the median, 25th and 75th quartiles, and the minimum and maximum interquartile intervals between concentrations. Radar graphs were employed to show the frequency (%) of the degree of reversibility between concentrations for gill and liver histopathology. The IBM SPSS Statistics version 22.0, GraphPad Prism 8.1 (San Diego, California, USA), and OriginPro version 2022 (OriginLab Corporation, Northampton, MA, USA) software packages were utilized for statistical analyses. A significance level of p < 0.05 was considered statistically significant.

Results

Nuclear cytotoxicity/genotoxicity and IOD

The response pattern of CL-T was concentration dependence. After 96 h, a significant increase in total NA was observed at a concentration of 200 mg/L compared to the control group (Fig. 1) (p = 0.014).

Figure 1 Nuclear abnormalities frequency verified in zebrafish adults exposed to 70, 140 and 200 mg/L of Chloramine-T for 96h.

Values are expressed as medians, 25 and 75th quartiles. Medians with different letters are significantly different. Distinct letters represent a significative difference p < 0.05; +, mean.

The MN assay showed no genotoxic effects of CL-T. However, IOD decreased at concentrations of 70, 140, and 200 mg/L. At 200 mg/L, the curve fit displayed a different profile compared to 70 and 140 mg/L (Fig. 2). Notably, the increase in IOD was only detected in a specific pixel wave (3,5–5,8) at a concentration of 200 mg/L.

Figure 2 Histograms representing the integrated optical density (IOD) from erythrocytes nuclei of zebrafish adults exposed to 70, 140 and 200 mg/L of Chloramine-T for 96 h.

AChE inhibition

During the 96-h exposure to sublethal concentrations of CL-T, AChE activity was not significantly inhibited in the brain of adult zebrafish (Fig. 3).

Figure 3 Acetylcholinesterase (AChE) activity observed in zebrafish adults exposed to 70, 140 and 200 mg/L of Chloramine-T for 96 h.

Values are expressed as medians, 25 and 75th quartiles for AChE activity in micromoles per minute per milligram of protein (µmols/min − mg protein). The same letters represent that there was no significant difference (p < 0.05) between the control and concentration groups (mg/L).

Gill and liver histopathology

Figure 4 illustrates the DTC in the gills and liver, along with the frequency of respective importance scores (w1, w2, and w3). Figures 5 and 6 depict the control and exposed specimens, respectively. All fish exposed to CL-T exhibited higher DTC. In the gills, although the concentrations differed from each other, the occurrence of w2 changes was approximately 50% across all concentrations. W3 changes were observed in 25% of the samples exposed to a concentration of 200 mg/L. The most frequently observed alterations included dilatation of lamellar capillaries, edema in the secondary lamellae (sl) (Fig. 5B), and adhesion and complete fusion of the secondary lamellae (Figs. 5C and 5D).

Figure 4 Degree of Tissues Changes (DTC) and frequency of w1, w2, and w3 reversibility degree of damage for the gills and livers in zebrafish adults exposed to 70, 140 and 200 mg/L of Chloramine-T for 96 h.

(A) and (B), box plot for the median, 25th, 75th quartiles and mean (+) degree of tissues changes (DTC). (C) and (D), radar graphic visualization for the w1, w2, and w3 frequency distribution for gill and liver, respectively. Distinct letters among boxes represent difference (P < 0.05).

Figure 5 Histological sections of gills stained in HE from zebrafish adults control and exposed to 70, 140 and 200 mg/L of Chloramine-T for 96 h.

(A) Control specimen on the sagittal section; primary lamella (pl) and secondary lamella (sl) appear intact morphology; er (erythrocytes); sinus venous (black star); bar scale = 50 µm; in highlighted, squamous epithelial cells (ep) form a thin lining layer along the primary lamella; chlorite cells (white arrow) and pillar cells (p) are usually observed; bar scale = 10 µm. (B) A transversal section of a specimen exposed to 70 mg/L; melanomacrophages (Mm) are readily aggregated peripherally to the arterioles; bar scale = 100 µm; in highlighted, an initial aneurism formation (arrow); bar scale = 10 µm. (C) A gill segment in a sagittal section of specimen exposed to 140 mg/L showing secondary lamellae fusioned (arrowhead) and focal epithelial cells necrosis; bar scale = 100 µm. (D), (E) and (F) Sagittal sections from specimens exposed to 200 mg/L showing distinct lamellar fusion morphology: bar scale = 50 µm. In (D), an initial lamellar fusion due to epithelial cells proliferation (white arrow); in (E), a thin layer of epithelial cells lines the fully fused lamellae (arrows); basophilic nuclei niches are diffusely distributed diffusely distributed according to the extensive cellular proliferation; and (F), layers of the same cells thicken the gill lining, suggesting a metaplasia adaptation (white arrow); arrowhead shows epithelial cells desquamation.

In the liver, a high DTC was observed at 200 mg/L. Furthermore, a w2 degree of reversibility was found in all concentrations, with a frequency of approximately 50%. Liver structural changes ranged from vascular hyperemia to focal and diffuse necrosis, although these occurred less frequently. Hepatocellular changes were characterized by hypertrophy, vacuolar degeneration (hydropic and fatty), and the presence of typical degenerative nuclear figures (Fig. 6).

Figure 6 Histological sections of liver parenchyma in zebrafish adults exposed to 70, 140 and 200 mg/L of Chloramine-T for 96 h.

(A)–(C) Control samples. (A) Low magnification showing a typical and homogeneous architecture; portal vessels (bv) are randomly distributed and partially filled by blood cells; sinusoids displayed narrow unfilled spaces which usually connect to portal vessels; bar scale = 100 µm. (B) Detail of intact sinusoidal web; some hepatocyte displayed normal cytoplasmatic vacuoles suggestive of lipids or glycogen stocks; a cordonal hepatocytes arrangement are commonly noted; bar scale = 50 µm. (C) Sinusoid aspect in high magnification; intra sinusoidal macrophage (M) are adhered to the endothelial wall; eosinophilic granular pigments are typical in the cytoplasm of hepatocytes which present nuclei basophilic with evident nucleoli; bar scale = 10 µm. (D) and (E) Sample from a specimen exposed to 70 mg/L. Sinusoidal web and portal vessels (pv) are mildly distended and filled by leucocyte (L) cells; hepatocytes cytoplasm showing enlarged aspect; bar scale = 50 µm; in (E), sinusoidal architecture is disrupted with lumen content serous exudate (E); hepatocytes are notably modified displayed increased cytoplasmic content suggestive of hydropic degeneration; bar scale = 50 µm. (F) Sample from a specimen exposed to 140 mg/L; general aspect showing hepatocytes clearly vacuolated, typical of fatty degeneration; portal vessel is distended and hyperemic; the sinusoidal web is not apparent due the hepatocellular compression; bar scale = 100 µm. (G) Sample from a specimen exposed to 200 mg/L; necrosis areas are diffusely distributed into the parenchyma, although some region the tissue architecture is preserved; bar scale = 150 µm. (H) and (I) high magnification of the 200 mg/L sample; cell necrosis (n) area surrounded by hepatocytes in degeneration process; nuclei are pyknotic (pk), in karyorrhexis (kh) or karyolisis (ka) stages; bar scale = 10 µm. HE stain.

Discussion

The presence of NA in our study indicates a possible cytogenotoxic effect of CL-T on adult zebrafish (Harabawy & Mosleh, 2014). NA have been widely used as biomarkers for cytogenotoxicity and are associated with the replication of damaged cells (Erbe et al., 2011). The significant increase in NA observed, particularly at the sublethal dose of 200 mg/L of CL-T, suggests a possible cytogenotoxic effect on adult zebrafish. Exposure to xenobiotics at different stages of cell division can result in nuclear protrusions or invaginations in erythrocytes, leading to impaired protein formation (Fasulo et al., 2010; Mitchelmore & Chipman, 1998). Additionally, DNA damage can be heritable, causing mutations and cell proliferation (Francisco et al., 2019).

Our findings demonstrate that CL-T at a concentration of 200 mg/L has significant effects on zebrafish erythrocyte DNA. Disinfectants used in water disinfection, such as chlorine, ozone, UV, and chloramines, can react with natural organics to produce trihalomethanes and disinfection byproducts (DBPs), which have genotoxic effects and potential carcinogenic properties (Gustavino et al., 2005). The cytogenotoxic effects of CL-T may be attributed to the cross-linking of DNA within and between strands, occurring when exogenous or endogenous substances react with two DNA nucleotides, forming covalent bonds. These adducts interfere with cell metabolism and can trigger cell death, while also providing insights into how proteins interact with DNA. Chloramines can be formed through the interaction of hypochlorous acid with DNA bases, subsequently decomposing into aminyl radicals. N-radicals located at the exocyclic amino positions of cytosine and adenine are the major radical adducts of a nucleoside mixture (Cadet & Wagner, 2013; Gustavino et al., 2005).

Although our results did not show a significant increase in MN frequency in fish erythrocytes exposed to CL-T for 96 h, MN formation typically arises from chromosomal breaks or mitotic anomalies, which require the occurrence of the mitosis process (Shi et al., 2009). However, CL-T was observed to affect the chromatin of zebrafish peripheral erythrocytes, as indicated by the increase in nuclear density observed in the IOD analysis. This effect of CL-T could influence chromatin condensation, organization, and positioning, as well as mitotic spindles (Hübner & Spector, 2010). Variations in the density of chromatin attachment sites could explain the observed differences in chromatin mobility during the cell cycle and cellular development (Vazquez, Belmont & Sedat, 2001). While major chromosomal rearrangements or translational mobility at the level of individual chromosomes are not apparent during interphase, chromatin dynamics are rapid enough to allow intrachromosomal interactions, such as the cis or trans association of an enhancer and a promoter, occurring within seconds and spanning distances of less than 1 µm (Hübner & Spector, 2010). Analysis of erythrocyte maturation stages in Cyprinus carpio revealed low- and high-density chromatin domains, as well as an interchromatin domain with high light transmittance and low optical density. CL-T could not only alter chromatin organization and condensation but also affect the cell cycle, leading to the detection of mature erythrocytes only after 96 h of exposure (Rothmann et al., 2000).

AChE, EC 3.1.1.7 is a crucial enzyme in the nervous system that is responsible for terminating nerve impulses through the hydrolysis of the neurotransmitter acetylcholine (ACh). It plays a vital role in cholinergic neurotransmission processes (Lionetto et al., 2013; Rang et al., 2016). Previous studies have shown significant AChE inhibition in embryos exposed to CL-T concentrations of 64 and 128 mg/L (Rivero-Wendt et al., 2023). Additionally, considering that AChE expression starts early, before synapse formation, and increases with embryo age (Teixidó et al., 2013), the seemingly contradictory results may suggest that embryos are more sensitive to the effects of CL-T compared to adults. A similar effect was confirmed in a study conducted with cetylpyridinium chloride on juveniles and adult zebrafish, where the impact of the product on neurotransmitters (serotonin, dopamine, and acetylcholine) was less in adults than in juveniles of the species (Dong et al., 2022). Although our results did not indicate significant differences in AChE inhibition, it is prudent to note that the evaluation of only one enzyme involved in neurotransmission processes, such as AChE, cannot rule out the neurotoxicity of CL-T.

CL-T induced an acute irritant response in gill mucosal and epithelial cells, consistent with previous reports (Quezada-Rodriguez et al., 2022). Teleost fish gills consist of four pairs of gill arches, each containing numerous filaments, and each filament is comprised of multiple folded lamellae forming a complex cellular layer. Gills are in direct contact with the environment and are vulnerable to morphological changes when exposed to substances that harm their tissues. Toxic interactions with different stages of branchial transport or infusion patterns can significantly affect the fish’s ability to regulate osmotic balance in fresh water environments (Evans, 1987; Fernandes et al. 2020; Tavares-Dias, 2021).

In the present study, the impact of CL-T concentration on DTC in the gills varied according to the concentration. However, all concentrations showed significant effects compared to the control. W2 lesions were the most prevalent, characterized by rupture of the lamellar epithelium and lamellar thrombosis. Previous research has highlighted the effects of various drugs on gill morphology in teleosts. For instance, exposure to Halamid® at concentrations of 15–200 mg/L for 96 h in Danio rerio resulted in changes such as congestion, edema, epithelial detachment, hyperplasia, cellular hypertrophy, telangiectasia, and necrosis of the respiratory epithelium (Alidadi Soleiman et al., 2017). Similarly, in Atlantic salmon (Salmo salar), acute exposure to Halamid® (CL-T) at concentrations of 25 and 50 mg/L for 12 h led to desquamation of epithelial cells, multifocal hyperemia, and extensive epithelial necrosis in the primary and secondary lamellar segments (Powell & Harris, 2004). In juveniles of Arapaima gigas, concentrations ranging from 50 to 100 mg/L resulted in hyperplasia, sinus dilation, epithelial detachment, and lamellar fusion, which are nonspecific defense mechanisms in the presence of irritants (Cordeiro Bentes et al., 2022). Our findings demonstrate that the histopathological effects observed in the gills were reversible, characterized by epithelial responses without progression to necrosis or structural dissociative processes. In contrast to the gill histopathological findings, the liver showed higher frequencies classified as w3. The liver’s metabolism and morphology are known to be sensitive to various environmental toxins (Gu & Manautou, 2012). Reversible changes such as hydropic swelling, atypical vacuoles, hepatocellular lipidosis, and irreversible changes were observed. Focal and diffuse necrotic areas were more common in specimens exposed to 200 mg/L of CL-T. Additionally, degenerative nuclear figures were frequently observed at concentrations of 140 and 200 mg/L. Limited data are available regarding safe treatment concentrations of CL-T for freshwater fish. Baths of 180 min at concentrations of 20, 50, and 80 mg/L of CL-T were insufficient to cause liver damage in Ictalurus punctalus (Gaikowski, Densmore & Blazer, 2009). Conversely, our results suggest that liver injury following CL-T exposure may be attributed to elevated levels of oxidative stress, indicating disruption of homeostatic and functional membrane disruption (Bilzer & Lauterburg, 1991; Tatsumi & Fliss, 1994; Tkachenko, Kurhaluk & Grudniewska, 2013).

Conclusions

Overall, our results indicate that changes in the erythrocyte nuclear chromatin condensation profile can be observed in association with branchial and hepatic toxicity, as determined by NA and IOD measurements. In the case of short-term exposure, tissue changes suggest low to moderate toxicity responses for adult samples. In the gills, nonspecific reversible lesions can be observed, whereas in the liver, irreversible lesions with a more pronounced harmful potential are present.

Supplemental Information

Supplemental Information 1 Statistical analysis - data

Click here for additional data file.

Supplemental Information 2 21 points checklist

Click here for additional data file.

Supplemental Information 3 Smears preparations and respective images digitally processed for the density optical integrated analysis

(A) a control sample; the arrow shows a regular shape and intact erythrocyte nucleus. (B) respective image in threshold adjusts; arrow shows the same nucleus from image a; yellow contour delimits the nuclear area for the pixel measure; (C) and (D) a Chloramine T (200 mg/L) group sample; note the abnormal shape and colorless nuclei (arrowhead) representative of irregular chromatin condensation. Bar scale = 10 µm.

Click here for additional data file.

Supplemental Information 4 Data: normalized Density Optical Integrated

Click here for additional data file.

Additional Information and Declarations

Competing Interests

Author Contributions

Animal Ethics

Data Availability

Carlos Eurico Fernandes is an Academic Editor for PeerJ.

Carla Letícia Gediel Rivero-Wendt conceived and designed the experiments, performed the experiments, analyzed the data, prepared figures and/or tables, authored or reviewed drafts of the article, and approved the final draft.

Ana Luisa Miranda Vilela analyzed the data, authored or reviewed drafts of the article, and approved the final draft.

Luana Garcia Fernandes performed the experiments, analyzed the data, prepared figures and/or tables, authored or reviewed drafts of the article, and approved the final draft.

Andreza Negreli Santos performed the experiments, analyzed the data, prepared figures and/or tables, authored or reviewed drafts of the article, and approved the final draft.

Igor Leal performed the experiments, analyzed the data, prepared figures and/or tables, authored or reviewed drafts of the article, and approved the final draft.

Jeandre Jaques conceived and designed the experiments, performed the experiments, authored or reviewed drafts of the article, and approved the final draft.

Carlos Eurico Fernandes conceived and designed the experiments, performed the experiments, analyzed the data, prepared figures and/or tables, authored or reviewed drafts of the article, and approved the final draft.

The following information was supplied relating to ethical approvals (i.e., approving body and any reference numbers):

Ethics Committee for the Experimental Use of Animals (CEUA; approval no. 3.128/2021).

The following information was supplied regarding data availability:

The raw measurements are available in the Supplementary Files.

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
