# Peer review of "Cytogenotoxic potential and toxicity in adult Danio rerio (zebrafish) exposed to chloramine T"

_PeerJ, doi:10.7717/peerj.16452_

## Round 0.1 · original submission · Major Revisions

The three reviewers unanimously appreciate the environmental relevance and importance of your study, but still see room for improvement, especially in the introduction (referring to previous studies, see reviewer 2), the reporting of results, but also with regard to the statistical analysis (see reviewer 1). Also reviewers 1 and 2 identified inconsistencies regarding the research question, which is currently changing in the course of the manuscript.

I look forward to your revised manuscript, ideally incorporating the reviewers' suggested changes.

Reviewer 1 ·

Basic reporting

no comment

Experimental design

no comment

Validity of the findings

no comment

Additional comments

The study provides interesting results about the effects of chloramin T on different endpoints of adult zebrafish after 96 h exposure. The findings of this study are environmentally relevant in the context of disinfection use in aquaculture. While the study seems thoroughly conducted, the reporting of the results should be improved to allow the reader to understand and interpret the results. A detailed feedback on the different paragraphs can be found below:

Introduction:
The introduction is well written and provides all necessary background to the reader. There are multiple inconsistencies with abbreviations. Once an abbreviation is introduced, it should be used consistently (applies to all paragraphs).

Material and methods:
Very detailed. All information provided.
Statistics: it is not entirely clear how the data was handled regarding individuals and replicate treatments. Were the mean values of the triplicates used or the ones of all individuals from one treatment?

Results:
A paragraph about survival and behavior of the fish is missing. From M&M it seems that some fish died.
The results paragraph needs major revision. The structure is confusing and it is very superficially written, without explaining the observations. Paragraphs about different endpoints are not correctly divided. The first paragraph mentions histology, but does not report about it. The description of results is often imprecise. “Significant difference” of a parameter is reported, but what is it exactly? Increase? Decrease? For example, in the IOD description it remains unclear what the changes in wave length actually mean.

Specifically:
Lines 216 – 218: the two sentences are contradictive. Was there another analysis earlier? And this contradicts the results shown in fig. 1, as in M&M a p-value of < 0.05 is defined as significant.

219-224: what is difference between wave length and wave pixel? What kind of changes are observed?

Line 240: it is not clear from what it can be concluded that reversibility may occur in this study. Fish were continuously exposed and no recovery phase was included in the experiment. Any statement about reversibility is speculative.

Figures:
Figure captions are partly superficial and missing details. Number of fish and replicates and replicates per endpoint not mentioned. Figures differ in style (labelling of x-axis). Figure 5: y-axis and w1, w2, w3 not defined.

Discussion:
The discussion provides a detailed analysis of all investigated parameters and compares to the available literature.

259: the concentrations used in this study induced lethality at a much lower concentration than in present study. This should be discussed. Mortality data should be reported.

316: please explain what is Halamid. Is it comparable to CL-T?

325: as mentioned above, this study cannot provide any data to conclude about reversibility of effects.

Conclusions:
This paragraph does not really fit with the aims and hypotheses stated in the introduction. Was the aim of this study to establish new biomarkers? From the introduction, it appeared that the effect of CL-T on fish and environment were not clear and should be evaluated in this study.

Reviewer 2 ·

Basic reporting

The intoduction provides a good background of the study, and refers to earlier studies in the same test organism at a different developmental stage. However, no details of that study are provided, and thus no support for the present study is created. I have advised a change of the paragraph in question.
Figure 2 is not mentioned in the results other than as a support for Figure 3. Otherwise the figures are clear and of high resolution.

Experimental design

The research question is well defined and the manuscript states knowledge gaps to be filled. However, the question seems to change throughout the manuscript, and the conclusion makes no mention of any of the aims.
The method, although thought through, is not well detailed in some aspects. These have been highlighted in the review. As it is now, the method would not be suitable for replication by other researchers.

Validity of the findings

The conclusions do not refer to the aims of the study. The manuscript states that the aim is to assess the lethal and sublethal impact of CL-T on the zebrafish using various methods. Not only was there not a single mention throughout the document of lethal effects, or a tabulation of EC and LC values, but also the conclusions states that the results demonstrate the use of some of the methods as biomarkers.

Annotated reviews are not available for download in order to protect the identity of reviewers who chose to remain anonymous.

Reviewer 3 ·

Basic reporting

In this study, the cytotoxic and genotoxic effects of Chloramine-T, which is used as a disinfectant in aquaculture, on the aquatic vertebrate model organism zebrafish were investigated. The language of the article is clear and fluent. In addition, the material and method are explained in detail. Similarly, the findings are described in detail and the tables and figures are well-illustrated. It is appropriate to publish the article in this form.

Experimental design

In the materials and methods section, the procurement of the organism, its care under laboratory conditions, and the ethical committee of the study are given. In addition, chemical exposure and subsequent experimental phases were explained in detail. Thus, it can be a resource for researchers who want to work with similar methods.

Validity of the findings

The findings of the study were given in detail in the text as well as in tables and figures.

Additional comments

No comments.

---

## Round 0.2 · Major Revisions

Thank you very much for the revision which has led to a significant improvement of the manuscript. However, two key aspects are still not addressed although clearly highlighted by the reviewers in the first revision round: (1) mortality data and (2) sex ratio during the 96 h exposure experiment. Both issues have to be resolved as a requirement for a final acceptance of the manuscript. In the annotated pdf file attached, I have also commented on some minor issues and proposed corrections (highlighted by yollow background) that you should also take into account.

Reviewer 1 ·

Basic reporting

ok

Experimental design

ok

Validity of the findings

ok

Additional comments

The authors have sufficiently addressed almost all issue raised by the reviewers. The quality of the manuscript has clearly improved and the study could be published as it is. However, there is one important comment that the authors have not addressed: it is still not clear how many of the fish survived. I suggested in my previous review to add a paragraph about survival and behaviour observations to clarify this. The authors state in their rebuttal that they have addressed this point, but it is not in the manuscript. In line 149, the authors write "from individuals that survived". This implies that some did not survive. For data interpretation, is is crucial to know how many died in which concentration. In case this is misleading and 100% survived, this is also important information to add. Once this information is added, the manuscript can be accepted.

Reviewer 2 ·

Basic reporting

no comment

Experimental design

I question to suitability of the method. During the review it was made clear that the male:female ratio was not noted or taken into account, although the inherent biological differences can significantly impact toxicity. Further, rather than addressing the observed mortality in exposure groups (i.e., providing percentage mortality of control and treatment groups to determine potential statistical implications), the sentences eluding to these issues were just deleted. That does not answer the questions posed.
Lastly, none of the reasoning given in the response to reviewers for concentration selection were suitable, nor were they put into the manuscript. Selecting exposure concentrations for adult animals based on embryo data is not suitable, and explains why mortality was observed. The ranges must be tested in house, if published data is not found suitable.

Validity of the findings

Once the above comments are addressed, no comment.

---

## Round 0.3 · accepted · Accept

Thank you for the revision of the manuscript. I hereby certify that you have adequately taken into account all of the reviewers' comments, as I have checked by my own assessment of your revised manuscript. Based on my assessment as an Academic Editor, your manuscript is now ready for publication.